# Ketogenic Diet as a Nutritional Metabolic Intervention for Obsessive–Compulsive Disorder: A Narrative Review

**DOI:** 10.3390/nu17010031

**Published:** 2024-12-25

**Authors:** Astrid Lounici, Ana Iacob, Katarzyna Hongler, Melina A. Mölling, Maria Drechsler, Luca Hersberger, Shebani Sethi, Undine E. Lang, Timur Liwinski

**Affiliations:** 1Clinic for Adults, University Psychiatric Clinics Basel, University of Basel, 4031 Basel, Switzerland; astrid.lounici@upk.ch (A.L.); katarzyna.hongler@upk.ch (K.H.); undine.lang@upk.ch (U.E.L.); 2Pôle de Psychiatrie et Psychothérapie (PPP), Unité de Psychiatrie de Liaison, Hôpital du Valais, 1950 Sion, Switzerland; ana.iacob@hopitalvs.ch; 3MEDIAN Zentrum für Verhaltensmedizin, 31812 Bad Pyrmont, Germany; melina.anastasia.moelling@gmail.com; 4Stiftung für Ganzheitliche Medizin (SGM), Klinik SGM Langenthal, 4900 Langenthal, Switzerland; maria.drechsler@klinik-sgm.ch (M.D.); luca.hersberger@klinik-sgm.ch (L.H.); 5Metabolic Psychiatry, Department of Psychiatry and Behavioral Sciences, Stanford University School of Medicine, Palo Alto, CA 94305, USA; shebanis@stanford.edu

**Keywords:** ketogenic diet, obsessive–compulsive disorder, metabolic syndrome, metabolic psychiatry, nutritional therapy

## Abstract

The substantial evidence supporting the ketogenic diet (KD) in epilepsy management has spurred research into its effects on other neurological and psychiatric conditions. Despite differences in characteristics, symptoms, and underlying mechanisms, these conditions share common pathways that the KD may influence. The KD reverses metabolic dysfunction. Moreover, it has been shown to support neuroprotection through mechanisms such as neuronal energy support, inflammation reduction, amelioration of oxidative stress, and reversing mitochondrial dysfunction. The adequate intake of dietary nutrients is essential for maintaining normal brain functions, and strong evidence supports the role of nutrition in the treatment and prevention of many psychiatric and neurological disorders. Obsessive–compulsive disorder (OCD) is a neuropsychiatric condition marked by persistent, distressing thoughts or impulses (obsessions) and repetitive behaviors performed in response to these obsessions (compulsions). Recent studies have increasingly examined the role of nutrition and metabolic disorders in OCD. This narrative review examines current evidence on the potential role of the KD in the treatment of OCD. We explore research on the KD’s effects on psychiatric disorders to assess its potential relevance for OCD treatment. Additionally, we identify key gaps in the preclinical and clinical research that warrant further study in applying the KD as a metabolic therapy for OCD.

## 1. Introduction

Obsessive–compulsive disease (OCD) is a mental disorder characterized by obsessions, such as intrusive thoughts, and compulsions, such as repetitive behaviors, which are often triggered by these thoughts and serve as attempts to alleviate anxiety or heightened inner tension or to achieve a sense of “completeness” [1]. According to the World Health Organization WHO, OCD is amongst the top ten most disabling illnesses [2]. While subthreshold symptoms affect around a third of the population [3], approximately 1–3% of individuals are affected globally by a diagnosed case of OCD over a lifetime [4,5]. Around 10% of individuals with OCD are treatment-resistant despite adequate treatment [6], and augmentation with psychotropic medication for those who do not respond sufficiently to psychotherapy can be associated with a substantial burden of side effects [7]. OCD often begins early and is long-lasting, with nearly 25% of males in the National Comorbidity Survey Replication (NCS-R) reporting onset before age 10 [8]. In females, OCD often begins in adolescence but may emerge in the peripartum or postpartum period [9]. Research on metabolism and nutrition in OCD remains limited, but the growing interest in metabolic and nutritional interventions in psychiatry warrants evaluating the existing literature to determine the potential of such approaches in OCD, as has been explored for bipolar disorder (BD) and schizophrenia (SZ) [10,11].

This review highlights metabolic and related alterations in OCD that can be targeted by lifestyle interventions, with a focus on well-supported changes identified through high-quality studies. It emphasizes the ketogenic diet (KD) as a promising intervention in psychiatry [10,12,13,14,15], supported by evidence linking diverse physiological alterations to its multipronged effects [16]. The evidence is synthesized to provide a strong rationale for studying the KD in OCD, encouraging further research and clinical trials in this critical area with significant unmet needs.

## 2. Method

This narrative review examines the relationships between OCD, insulin signaling, diabetes, glucose metabolism, metabolic syndrome, and the KD. A comprehensive search was performed using MEDLINE via PubMed and Embase. The search terms included “obsessive–compulsive disorder”, “obsessions”, “compulsions”, “diabetes”, “insulin”, “insulin signaling”, “glucose metabolism”, “metabolic disorders”, “metabolic syndrome”, and “ketogenic diet”, combined using Boolean operators. Peer-reviewed English-language studies were included without date restrictions. This review analyzes current evidence and concludes with considerations for future research.

## 3. OCD Symptomatology, Diagnostics, and Current Treatment

OCD symptoms are characterized by the presence of obsessions and/or compulsions. Obsessions are repetitive, persistent thoughts, images, impulses, or urges that are intrusive and unwanted, often linked to significant anxiety. Compulsions, on the other hand, are repetitive behaviors or mental actions performed in response to obsessions according to rigid rules or in a specific manner to prevent a feared event or situation [17]. Children may struggle to identify obsessions, whereas most adults recognize both obsessions and compulsions [18]. The most common obsessive thoughts are contamination fears, pathological doubts, somatic obsessions, and an excessive need for symmetry. The most common compulsive behaviors are checking rituals, washing rituals, counting compulsions, and compulsive questioning [19]. The Dimensional Yale–Brown Obsessive–Compulsive Scale (Y-BOCS) categorizes obsessions and compulsions into seven dimensions, as described in Table 1 [20].

OCD primarily consists of both obsessive thoughts and actions. Purely one-sided manifestations are rare, while a combination of both happens in 90–95% [21]. Clinical and community studies show similar OCD features. Clinical studies identify symptom dimensions such as contamination (cleaning), harm (checking), and symmetry (ordering) [22,23]. Community surveys worldwide reveal similar OCD symptom profiles [8,24].

OCD exists on a spectrum alongside other disorders, such as pathological hoarding, hypochondria, Tourette syndrome, and body dysmorphic disorder. These disorders share common features, including the need for differential diagnosis. One alternative differential diagnosis is SZ [17]. In contrast to OCD, the obsessive thoughts in SZ may be experienced as auditory hallucinations or perceived as being imposed from outside [25]. In OCD, the person is aware that the intrusive thoughts are their own.

Onset often occurs around the age of 20 [8,26], while cases after the age of 30 are relatively rare [8]. Therefore, in childhood and adolescence onset, or onset after the age of 50, a physical cause must be considered [17,27]. After the age of 50, an organic brain-related evaluation should be conducted, and in children and adolescents, Pediatric Autoimmune Neuropsychiatric Disorders Associated with Streptococcal infections (PANDAS) can occur following a streptococcal infection [28]. Substance or medication-induced obsessive–compulsive symptoms (OCS) also represent an important differential diagnosis to OCD [29].

Significant stigma and feelings of shame are commonly associated with OCD [30], and individuals often deny or downplay their problems [31], which can lead to a correct diagnosis being delayed by many years [32]. Individuals with compulsive washing behaviors are frequently discovered only during dermatology consultations [33]. OCD is associated with high psychiatric and somatic comorbidity. Overall, 90% of individuals with OCD also develop another mental disorder [8], often affective and anxiety disorders [34], and suicidal tendencies are common [26]. Over 10% of individuals with OCD attempt suicide during their lifetime, while nearly 50% experience suicidal ideation [35]. Physical complaints, such as headaches and gastrointestinal issues, are also common in individuals with OCD [36].

Treatment options for OCD include cognitive–behavioral therapy (CBT) and augmentation with selective serotonin reuptake inhibitors (SSRIs), clomipramine, or antipsychotics [17]. CBT with exposure and response prevention has the strongest evidence of efficacy [37] and is superior to medication alone [38]. In general CBT for OCD shows the highest effectiveness among all psychotherapeutic treatments for major mental disorders [39]. Nevertheless, in only 50% of patients, a significant clinical improvement or remission of symptoms in response to CBT has been observed [40]. In cases of chronic progression and lack of response to CBT, medication should be considered. SSRIs are the first choice in pharmacological therapy, while clomipramine is the second choice due to its higher occurrence of side effects [17]. Evidence suggests that OCD drug response is better at higher doses than those typically used for other indications [41]. A network meta-analysis demonstrated that SSRIs are effective, with no significant differences in efficacy between agents in this class and comparable efficacy to clomipramine [42]. A combination of CBT with psychiatric medication seems to be more effective than CBT alone, at least in severe cases [42]. If the side effects allow, a higher dosage of SSRIs should be chosen, as this results in a greater effect [43]. If antidepressants show insufficient effectiveness, augmentation with atypical antipsychotics like risperidone and aripiprazole is an option as these agents have the best evidence [44]. Nevertheless, the effect size is moderate, and their use must be weighed against the side effects [17]. Both inpatient and outpatient therapies have proven to be effective. In cases where therapy slots are unavailable, internet-based therapy and self-help programs have also shown effectiveness and can be recommended during a waiting period [45].

Unfortunately, many OCD patients exhibit partial or poor responses to therapeutic interventions [46]. A treatment response is generally defined as a ≥25% reduction in Y-BOCS score after treatment initiation [47]. Early meta-analyses indicated that both SSRIs and clomipramine were more effective than a placebo, with clomipramine generally showing greater effect sizes [48]. Nevertheless, direct comparison studies have demonstrated comparable efficacy between clomipramine and SSRIs [48]. This may be due to earlier clomipramine studies involving treatment-naïve patients with low placebo response, whereas SSRI studies frequently included patients with a history of previous treatment failure with a higher placebo response. A Cochrane review calculated the number needed to treat (NNT) for SSRI treatment in OCD assuming a 10–20% expected response rate without treatment. If 10% of patients recover without treatment, twelve patients would need SSRIs to benefit one additional patient; if 20% recover without treatment, six patients would need SSRIs for one additional benefit [47].

It is therefore essential to enhance the efficacy of the existing treatments by incorporating multipronged strategies designed to address biological pathways that remain insufficiently targeted by current psychopharmacological and psychotherapeutic interventions.

## 4. Etiology of OCD

The etiology of OCD is multifactorial, consisting of biological, psychological, and external factors, and in 50–70% of cases, life events or stressors can be identified as triggers [49]. Family genetic findings indicate a prevalence of around 11% among relatives of OCD patients [50]. First-degree relatives have a 4.6 to 5.0 times higher risk of developing the disorder [51]. Genetic causes are also supported by twin studies, which show concordance rates of around 47% for dizygotic twins and up to 87% for monozygotic twins. The overall heritability estimate for OCD is approximately 48%, indicating a significant genetic contribution to the disorder [52]. Certain OCD subtypes, such as early-onset OCD with tics, may exhibit higher heritability [53]. Environmental factors, such as adverse perinatal events and stressful or traumatic experiences, have been identified as potential OCD risk factors [54,55,56]. Further research is needed to evaluate the relationship between the environment and OCD [18]. The role of lifestyle factors, such as diet, sleep, and exercise, is insufficiently understood [57,58].

Molecular genetic findings show that OCD is associated with changes in the serotonin and catecholamine system [59]. The glutamate system also seems to be affected, and a genome-wide association study (GWAS) involving 2688 OCD patients confirmed evidence for the involvement of the glutamatergic system, including the NMDA receptor gene [60]. Studies on copy number variants revealed a 3.3-fold increase in large deletions in OCD patients, which were linked to other neurodevelopmental disorders [61]. Finally, a comprehensive meta-analysis, which included 232,964 cases across various psychiatric disorders, found that OCD exhibited genetic correlations with anorexia nervosa, Tourette syndrome, BD, SZ, and depressive disorders, listed in decreasing order of frequency [62]. While the efficacy of SSRIs in OCD patients initially seemed to support the “serotonin hypothesis” of OCD [63], there is notably limited evidence supporting the idea that an underlying serotonin deficit plays a primary causal role in OCD [64]. Altered serotonin transporter receptor binding in regions like the midbrain has been observed in some studies, although the data are not entirely consistent [65,66]. Data on the dopaminergic system in OCD are conflicting [18].

Over the past three decades, human neuroimaging studies have consistently demonstrated abnormalities in the structure and function of cortical–basal ganglia–thalamic loops in individuals with OCD. Key findings include impairments in the orbitofrontal cortex (OFC) and the basal ganglia (specifically the caudate nucleus), suggesting disturbances in the cortico-striato-thalamo-cortical (CSTC) circuits [67]. Recent concepts also point to reduced top-down control due to cortical inhibitory mechanisms [68,69]. A worldwide “meta- and mega-analysis” by Boedhoe et al. indicated that OCD is linked to smaller hippocampal volumes and enlarged pallidal volumes [70]. Functional imaging has revealed reduced activation during inhibition tasks and decreased functional connectivity between key regions during cognitive tasks [71,72]. As a limitation, it should be noted that neuroimaging measures were no more effective than chance at identifying OCD, as indicated by a large machine learning study [73]. As a consequence of these brain changes, neuropsychological functions are expected to be impaired, which was confirmed in a meta-analysis showing moderate deficits in attention, executive functions (planning, flexibility, response inhibition), processing speed, and non-verbal memory [74]. While much translational research has focused on striatal abnormalities, findings in prefrontal cortical (PFC) networks, particularly within the OFC, have been especially consistent across studies [75]. Although less extensively studied, similar findings have been observed in other prefrontal cortical regions. Notably, significant evidence suggests that the anterior cingulate cortex (ACC) plays a key role in the generation of OCD symptoms [75]. Accumulated data show consistent hypoactivity in PFC regions during neurocognitive tasks, supporting the idea that this hypoactivity may be a cause, rather than a consequence, of OCD symptoms. Theories consistent with this include deficits in the PFC cognitive control network (including ACC/pre-SMA (pre-supplementary motor area), dorsolateral prefrontal cortex (DLPFC), inferior frontal junction, anterior insular cortex, dorsal pre-motor cortex, and posterior parietal cortex) may directly lead to compulsive behaviors [76,77]. In humans and monkeys with prefrontal damage, stimulus-bound behaviors occur, where strong sensory stimulus–action associations trigger reflexive responses. This phenomenon is observed in OCD, where individuals often engage in harm avoidance strategies in response to triggers before consciously implementing response prevention strategies [78]. PFC hypoactivity could thus impair inhibitory control, leading to more automatic compulsive behaviors, especially under stress. Hypoactivity in the ACC/OFC may also explain the reduced capacity of individuals with OCD to integrate changing values and apply them to behavior selection in complex or emotionally stressful situations. This helps explain why exposures are easier in controlled clinical settings but harder in real-world environments with dynamic contingencies [79]. Given the ACC’s role in error detection, evaluation, and action selection, its hypofunction may impair updating of values and decision-making, leading to persistent maladaptive behaviors [80,81]. Additionally, hypofunction in the OFC and ACC may hinder the learning of new associations between cues, actions, and outcomes, undermining the effectiveness of exposure therapy. These findings suggest that shifting to alternative primary energy sources, such as ketone bodies, may help address cerebral dysmetabolism and, as a result, reduce symptom burden [13].

Glutamatergic neurons from the PFC play a key role in CSTC circuitry, projecting to the striatum. Studies of cerebrospinal fluid (CSF) and magnetic resonance spectroscopy suggest alterations in glutamatergic metabolites, indicating a potential role in OCD, though findings are not entirely consistent [82]. Variants in glutamatergic genes, such as SLC1A1 and GRIN2B, are associated with OCD. A meta-analysis of GWAS has also implicated several genes in the glutamatergic system, including GRID2 and DLGAP1 [60,83]. Alterations in glutamate signaling and transmission have been linked to compulsive behavior in two knockout mouse models. These models focused on the scaffold proteins disks large-associated protein 3 (DLGAP3, also known as SAPAP3) and SLITRK5, which are present in the postsynaptic membranes of neuronal synapses, primarily in the striatum [84]. The signal attenuation model is a model based on classical conditioning, where rats initially press a lever for a food reward. A light stimulus is presented to facilitate the extinction of the food–stimulus association, yet some rats persist in compulsive lever-pressing. Rats treated with D-cycloserine (DCS), a partial NMDA receptor agonist, exhibited reduced compulsive behavior compared to controls [85]. Repetitive digging and marble-burying behaviors are models used to study obsessive–compulsive and related symptoms. Mouse studies show that the NMDA antagonist amantadine inhibits marble-burying behavior without affecting motor movement [86]. These findings suggest that altered glutamatergic neurotransmission is implicated in OCD and that reducing glutamatergic excitation may offer a promising treatment approach. Two studies examining the CSF of individuals with OCD reported elevated glutamate concentrations [87,88]. Further support for the role of glutamatergic pathways comes from pharmacological trials. Potential glutamatergic agents include riluzole, memantine, NMDA-receptor antagonists (e.g., amantadine, ketamine), anticonvulsants with glutamatergic properties (e.g., topiramate, lamotrigine), N-acetylcysteine (NAC), and DCS [84,89]. Riluzole, a medication that inhibits synaptic glutamate release, modulates ion channels, and promotes glutamate uptake by astrocytes, has demonstrated efficacy in open-label trials, reducing OCD symptoms in more than half of the pediatric and adult participants studied [90,91]. Nonetheless, no randomized controlled trials (RCTs) investigating the efficacy of riluzole in OCD have been published. An open-label study suggested memantine may be a useful augmentation strategy in treatment-resistant OCD [92]. A further open-label study showed memantine had some efficacy in OCD, but not in generalized anxiety disorder [93]. A recent randomized controlled cross-over trial found that a single dose of ketamine improved symptoms in individuals not receiving other OCD treatments [94,95]. Smaller trials of additional antiglutamatergic agents reviewed by Kariuki-Nyuthe show promising preliminary results in OCD [96].

Neuroinflammation has become a key focus in biological psychiatry. Several studies found abnormal distribution of inflammatory mediators in OCD, though findings are inconsistent. A meta-analysis of 12 studies examining the association between OCD and circulating cytokines (tumor necrosis factor (TNF-α), interleukin (IL)-1β, IL-6) revealed the following: (i) reduced levels of IL-1β in OCD compared to healthy controls, (ii) elevated IL-6 levels in treatment-naïve OCD patients compared to those who had received treatment, and (iii) increased TNF-α levels in OCD patients with comorbid depression [97]. These abnormalities may develop early in children, characterized by an overexpansion of CD16+ monocytes, which establishes a pro-inflammatory state that promotes the excessive secretion of Th1 cytokines, including Granulocyte–Macrophage Colony Stimulating Factor (GM-CSF), IL-1β, IL-6, IL-8, and TNF-α [98,99]. Elevated IL-2 levels, a key regulator of the immune response, have also been observed in OCD patients [100,101]. In a recent RCT, naproxen, as an adjunct to fluoxetine, showed superior efficacy over a placebo in reducing obsession and total OCD symptoms, with good safety tolerability [102]. However, a more sophisticated, multipronged, anti-inflammatory treatment approach is clearly required. More advanced interventions targeting neuroinflammation could offer a promising therapeutic avenue for OCD.

## 5. Metabolic Syndrome in OCD

Metabolic syndrome (MetS) is characterized by an array of inter-related conditions indicating metabolic and cardiovascular health risks [103]. Various definitions of MetS exist, with an internationally used definition featuring two modifications (Table 2) [104]. MetS pathophysiology is complex and multifactorial, with insulin resistance recognized as a key promoter [103]. The risk of cardiovascular disease is significantly higher in individuals with MetS [105]. However, the predictive value of MetS for atherosclerotic complications or mortality does not surpass that of its individual components. In addition to the defining criteria, other conditions associated with MetS include hemostatic disturbances, chronic inflammation, hyperuricemia, insulin resistance, and microalbuminuria [106]. As concerns about MetS grow in the general population, researchers have increasingly focused on its prevalence in psychiatric disorders. Patients with BD, generalized anxiety disorder (GAD), major depressive disorder (MDD), and SZ have a higher prevalence of MetS compared to the general population [107,108,109,110,111]. Longitudinal population studies indicate nearly twice the risk of cardiovascular disease and type 2 diabetes in individuals with SZ, BD, or MDD compared to the general population, independent of psychotropic medication use [112]. Conversely, up to 17% of diabetic patients report moderate to severe depressive symptoms, with 10% meeting the criteria for MDD [113].

Metabolic disruptions like hyperglycemia, hypertriglyceridemia, and weight gain are prevalent among individuals with serious mental illness [114,115]. These disturbances substantially elevate the risk of developing obesity, type 2 diabetes, cardiovascular diseases, and other chronic health issues. Remarkably, nearly two-thirds of patients initially admitted to hospitals due to acute psychosis develop obesity within a 20-year follow-up period [116]. Undesirable side effects, particularly metabolic issues, lead approximately 74% of individuals to discontinue antipsychotic medication within 18 months. This discontinuation contributes significantly to elevated rates of hospitalization and relapse among affected individuals [117]. Metabolic alterations in OCD patients are not well studied. Lifestyle factors associated with OCD, such as reduced physical activity, disruption of circadian rhythms, and sleep disorders, may contribute to the development of metabolic disorders by negatively impacting glucose metabolism and insulin sensitivity [57,58], but they are insufficiently understood. In a cohort study, Isomura et al. examined 25,415 individuals with OCD, comparing them to 12 million members of the Swedish general population and to their unaffected siblings. The findings revealed that OCD was associated with a heightened risk of metabolic and cardiovascular disorders in both comparisons, with increases of 45% and 47%, respectively. The risk for MetS in OCD patients was not associated with the dosage or duration of SSRI use [118]. A follow-up study by the same group found that OCD was linked to an increased risk of various cardiovascular diseases (CVDs) (adjusted HR [hazard ratio] for any CVD = 1.25). The strongest associations were observed for venous thromboembolism (adjusted HR = 1.48) and heart failure (adjusted HR = 1.37). Comparisons with non-exposed full siblings showed similar results [119]. A cross-sectional Italian study by Albert et al. found that 21.2% of 104 OCD patients had MetS, and its presence was associated with smoking, low or no physical activity, higher BMI, and longer exposure to antipsychotics [120]. A subsequent Italian study found comparable rates of metabolic disorders in 162 OCD patients, with 19.8% meeting the criteria for MetS and 6.2% diagnosed with diabetes [121]. The presence of medical conditions was associated with older age and lack of physical activity. A more recent cross-sectional study of 107 OCD patients found a MetS prevalence of 39.2%. Abdominal obesity was the most common component (68.2%), followed by low HDL (high-density lipoprotein) cholesterol (50.5%). High serum triglycerides, fasting glucose, high systolic blood pressure, and high diastolic blood pressure were observed in 47.7%, 20.6%, 18.7%, and 9.3% of patients, respectively [122]. These findings align with a large epidemiological study in Singapore, which demonstrated that OCD patients had an elevated risk of diabetes compared to the general population, with an odds ratio of 3.1 [123]. Another Singapore survey, which analyzed changes in the comorbidity of mental and physical disorders between 2010 and 2016, found a significant increase in the prevalence of diabetes in OCD patients (from 4.1% in 2010 to 10.9% in 2016) and an increase in the prevalence of OCD in diabetic patients (from 1.4% in 2010 to 3.9% in 2016) [124]. A large study involving >9000 Chinese adults found a higher prevalence of obsessions in newly diagnosed diabetic and pre-diabetic individuals compared to non-diabetic controls, with odds ratios ranging from 1.20 to 1.29 [125]. A Brazilian study on insulin-dependent type 2 diabetic patients found a higher prevalence of anxiety disorders and OCD compared to healthy controls, with an odds ratio of 2.47 for OCD [126]. Two independent studies found a higher prevalence of OCS in uncontrolled compared to controlled diabetic patients, with a correlation between OCS and HbA1c levels. In the first study, the correlation was particularly pronounced in women. In the second study, uncontrolled diabetic patients, regardless of gender, had a 5.5-fold higher risk of OCS. Notably, the study reported that over 50% of the 400 T2D patients had OCD symptoms (Y-BOCS score > 15), with women having double the risk of OCS [127,128].

In summary, evidence shows that individuals with OCD face a heightened risk to their metabolic health, indicating the need for medical care alongside standard OCD treatments or a dual therapy approach targeting both mental and metabolic symptoms.

## 6. Molecular Metabolic Alterations in OCD

Insulin plays key non-metabolic roles in the brain, supporting neural survival, synaptic and dendritic plasticity, learning, memory, neural circuit formation, and modulation of dopaminergic transmission in the striatum and brainstem [129,130,131,132,133,134,135,136,137]. To exert these functions, insulin crosses the blood–brain barrier and binds to specific insulin receptors on neurons and glial cells [129]. Emerging evidence highlights brain insulin resistance as a potential factor in cognitive decline and a mediator linking early life adversity to physical and mental health outcomes [138,139]. Despite the growing literature on diabetes and mood disorders, the connection between diabetes, insulin signaling, and OCD remains poorly understood. Mediating factors in diabetes-related mood disorders, such as inflammation, hypothalamic–pituitary–adrenal (HPA) axis dysregulation, and gut microbiota alterations, have also been linked to OCD [140,141,142,143,144,145,146].

A shared genetic risk has been identified between OCD/OCS and insulin-related traits [147]. A GWAS conducted in the Philadelphia Neurodevelopmental Cohort (PNC) (650 children and adolescents) examined total OCS scores and six OCS factors derived from an exploratory factor analysis of 22 items. Validation in the independent Spit for Science cohort (5047 children and adolescents) revealed significant genetic overlap between OCD and “guilty taboo thoughts”. Additionally, genetic sharing was observed between “symmetry/counting/ordering” and “contamination/cleaning”. A central nervous system (CNS) insulin-linked gene set was associated with “symmetry/counting/ordering” in the PNC. Genetic sharing was also found in peripheral insulin signaling traits: type 2 diabetes with “aggressive taboo thoughts”, and fasting insulin and 2 h glucose levels with OCD. These findings suggest that OCD, OCS, and insulin-related traits share genetic risk factors, pointing to a common etiological mechanism underlying both somatic and psychiatric disorders [147].

TALLYHO/JngJ (TH) mice serve as a reliable model for diabetes, developing hyperglycemia, hyperinsulinemia, and enlarged islets of Langerhans [148]. In a recent study, TH mice exhibited increased compulsive-like and anxiety behaviors, as evidenced by reduced spontaneous alternation and time spent in the open arm [149]. These behaviors correlated with blood glucose levels. Magnetic resonance spectroscopy (MRS) showed elevated glucose levels in the dorsal–medial striatum (DMS), which were associated with compulsive-like behaviors. Diffusion tensor imaging (DTI) revealed reduced fractional anisotropy in the DMS, suggesting altered structural connectivity linked to behavioral inflexibility. This may indicate that increased glucose in the DMS contributes to compulsive-like behaviors via changes in brain connectivity. Additionally, TH mice showed higher glutathione levels in the anterior cingulate cortex, a region associated with OCD, and decreased insulin growth factor 1 (IGF-1) in the cerebellum, with increased IGF-1 levels in the blood [149]. A study showed that streptozotocin (STZ)-induced diabetic mice exhibited increased compulsive behaviors, as evidenced by enhanced marble burying compared to non-diabetic mice. Treatment with metformin and genistein, a neuroprotective and anti-inflammatory isoflavonoid with anti-diabetic properties, significantly reduced compulsive behaviors to levels similar to those seen with fluoxetine [150].

Future research using prospective designs with long-term follow-up is needed to better clarify the role of comorbidities, lifestyle habits, biological and genetic factors (e.g., inflammatory status and familial risk factors), and medications in the development of insulin-signaling dysregulation in OCD patients. Metabolic alterations at the molecular level and metabolic syndrome can be effectively addressed through lifestyle interventions, such as low-carbohydrate diets as well as the KD, intermittent fasting, and the Mediterranean diet [16,151,152,153,154,155,156,157,158]. However, no studies have analyzed the impact of lifestyle factors—such as daily physical activity, sleep patterns, and nutrition—on the metabolic profiles and mental symptoms of patients with OCD.

## 7. Mitochondrial Dysfunction in OCD

Increasing evidence suggests that dysfunctional mitochondria play a significant role in the development of psychiatric disorders. Genetic, postmortem, imaging, and induced-pluripotent stem cell (iPSC) studies of a range of psychiatric disorders have consistently highlighted the role of mitochondrial dysfunction in their common pathogenic mechanisms [159,160,161,162,163,164,165,166]. Mitochondria generate adenosine triphosphate (ATP) through oxidative phosphorylation (OXPHOS) via the electron transport chain (ETC), which consists of five complexes. This process is vital for the normal physiological function of neurons in the brain, including synaptic transmission [167,168,169,170]. Given the brain’s high energy demands for normal function, it is plausible that dysfunctional energy metabolism due to mitochondrial dysfunction may impair brain functions, potentially contributing to neuropsychiatric disorders [171].

Recent studies have highlighted the significant roles of oxidative stress, free radicals, inflammation, and mitochondrial dysfunction in the development of OCD [172,173]. Mitochondrial dysfunction and oxidative stress are closely interconnected, with each influencing the other. A positron emission tomography (PET) study revealed inflammation not only in the basal ganglia but also in the CSTC circuit in individuals with OCD [140]. A cross-sectional study revealed significantly elevated serum levels of 8-hydroxydeoxyguanosine, a marker of oxidative DNA damage, in individuals with OCD compared to those without the disorder. Notably, these levels were reduced in patients undergoing treatment for OCD [174].

Although cognitive impairment is common in psychiatric disorders, the link between psychiatric symptoms and mitochondrial disorders has been minimally studied. Current knowledge mainly stems from a limited number of case reports [175]. One study involving 36 patients with mitochondrial diseases (type unspecified) used a structured psychiatric assessment to evaluate psychiatric symptoms [176]. Mitochondrial diseases are increasingly recognized as common genetic disorders [177], but the prevalence of mitochondrial dysfunction in OCD or other anxiety disorders remains unclear. A systematic review and meta-analysis evaluated the efficacy, acceptability, and safety of mitochondrial modulators (eicosapentaenoic acid, folic acid, inositol, lithium, N-acetylcysteine, and silymarin) in comparison to conventional treatments for OCD and related disorders [178]. Mitochondrial modulators, as a group, significantly improved OCD and anxiety symptoms. Specifically, lithium and N-acetylcysteine notably reduced OCD symptoms, with N-acetylcysteine also slightly improving anxiety. Mitochondrial modulators marginally outperformed a placebo in improving Sheehan Disability Scale (SDS) scores and treatment response, with N-acetylcysteine showing particular benefit in both areas. Notably, N-acetylcysteine had no reported adverse events, making it a promising adjunctive antidepressant treatment for individuals with OCD and related disorders who have undergone conventional treatments [178].

## 8. The Ketogenic Diet and Ketone Body Metabolism

Ketogenesis is a phylogenetically old process characterized by increased levels of circulating ketone bodies. Ketone bodies are produced in a process termed ketogenesis, which occurs in the mitochondrial matrix of hepatocytes [179]. Ketone bodies comprise acetoacetate, 3-β-hydroxybutyrate (BHB), and acetone. Ketosis can be achieved by either fasting, prolonged exercise, or consuming a diet vigorously reduced in carbohydrates, usually <20–50 g of net carbs per day [180]. The KD aims to achieve ketosis by consuming food items balanced in protein, very low in carbohydrates (typically <20 g per day), and rich in fats (usually >70% fat) [181]. KD meals primarily consist of high-fat foods such as butter, cream, mayonnaise, oils, and protein sources like meat, fish, eggs, and cheese, with small to moderate portions of vegetables or salads and a minimal amount of fruit (mainly low-fructose berries) to drastically reduce carbohydrate intake [182]. The KD is tailored to each individual, with regular monitoring by clinicians and dietitians experienced in its use essential for tracking progress and managing potential side effects. The KD requires significant commitment, motivation, and support for successful adherence. However, increased educational resources, protocols, recipes, meal planning tools (including patient-friendly software), and support groups are improving the diet’s acceptability, accessibility, and convenience [182]. The average concentration of total ketone bodies in carbohydrate-consuming individuals is <0.03 mmol/L, showing a certain degree of circadian oscillation between 0.05 and 0.25 mmol/L. As per Stryer Biochemistry 2019, when blood ketone levels range from 0.3 to 0.5 mmol/L, the brain generates 3–5% of its ATP from ketones. At 1.5 mmol/L in the blood, this percentage can increase to up to 18%, and at 5 mmol/L, it can reach up to 60%. During prolonged fasting, it may go up to 75%. Around three to four days of carbohydrate abstinence are required to cause the brain and other body tissues to switch to ketone bodies instead of glucose as the primary fuel for energy production [183]. Ketosis represents a phylogenetically conserved metabolic state with a long-standing role in human evolutionary history, underscoring its significance as an adaptive mechanism. Mild ketosis is frequently observed in both mothers and infants during the third trimester of pregnancy and at the time of birth [184]. Extended periods of fasting and ketosis were typical throughout human evolution, particularly during the Paleolithic era when societies were organized into small groups of hunter–gatherers [185,186]. The use of the KD for therapeutic aims in modern medicine dates back to the 1920s [180]. A century ago, the KD was a standard of care in diabetes, used to prolong the life of children with type 1 diabetes and to control the symptoms of type 2 diabetes in adults. However, the discovery of insulin in the 1920s enabled people with diabetes to countervail hyperglycemia despite high-carbohydrate diets [185]. Nowadays, the KD is an established, effective nonpharmacologic treatment for drug-resistant epilepsy in both children and adults and, thus, has a firm place in contemporary epilepsy treatment guidelines [187].

## 9. The Ketogenic Diet in Psychiatry

While the predominant therapeutic model in psychiatry primarily centered around psychopharmacology has yielded some success in tackling the global challenges of mental health, it falls short of addressing the complexity of the ongoing mental health crisis [188]. Recent evidence underscores the significance of nutrition in the prevalence and onset of mental disorders, highlighting its equivalence in importance to other medical disciplines [189]. In recent times, there has been a shift in therapeutic attention towards understanding how nutrition impacts the stability of neural networks in the brain, the levels of brain-derived neurotrophic factor, the functioning of ATP energy production, and the balance of neurotransmitters [190,191]. There is a growing consensus that cerebral glucose hypometabolism, insulin resistance, imbalances in neurotransmitters, mitochondrial dysfunction, oxidative stress, and inflammation play transdiagnostic roles in various neuropsychiatric disorders, such as Parkinson’s disease (PD), epilepsy, BD, SZ, and MDD [12,192]. The significant shortcomings of psychopharmacological treatments underscore the critical need for exploring new approaches to managing mental illnesses. One such intervention gaining traction in recent years is the KD. By limiting carbohydrate intake and inducing lipolysis, the KD prompts the production of circulating ketone bodies. These ketone bodies serve as an additional source of fuel for the brain, thereby reducing its reliance on glucose [193]. Furthermore, ketone bodies offer a multitude of effects that can be therapeutically leveraged, including the amelioration of metabolic dysfunction, such as improvement in the lipid profile and stabilization of insulin levels, inhibition of the mTOR (mammalian target of rapamycin) signaling pathway, enhancement in mitochondrial function and energy production, reduction in oxidative stress and inflammation, and rebalancing of the inhibitory–excitatory balance in the brain [13,16,194]. The efficacy of the existing psychiatric classification system, rooted in categorical diagnoses outlined in the ICD/DSM, continues to be subject to scrutiny. An encouraging alternative has emerged in the form of the “transdiagnostic” approach. This approach is anticipated to transcend conventional categorical diagnoses and expand upon them, aiming to enhance the classification and treatment of mental disorders [195,196]. There is a burgeoning interest within the psychiatric research community in utilizing personalized, transdiagnostic, dynamical systems to comprehend, model, diagnose, and treat psychopathology [197]. Growing evidence indicates a variety of common biological factors in mental disorders that could potentially be addressed through a personalized KD approach [198].

The potential of a low carbohydrate diet and the KD to reverse metabolic syndrome and metabolic dysfunction has been demonstrated by numerous randomized controlled trials [156,199,200,201,202,203,204,205,206,207]. Lately, there has been a notable increase in interest in conducting clinical trials across various psychiatric conditions to explore the potential of utilizing the KD as an adjunctive treatment [11,182,208,209,210]. As supported by a Cochrane review of randomized controlled trials and nearly a century of clinical application, the KD emerges as a safe and efficacious therapy for reducing seizures in children with drug-resistant epilepsy [211]. Recent narrative reviews suggest that the KD could influence the metabolic and biochemical aspects of serious mental illnesses. These potential effects encompass decreased oxidative stress, enhanced mitochondrial function and production, improved glutamate/GABA (γ-aminobutyric acid) signaling, and lowered levels of intracellular sodium and calcium [13,212]. In a study involving 28 treatment-resistant patients with MDD, BD, and SZ, the KD resulted in a notable enhancement in psychiatric symptoms in 100% of the patients. Clinical remission was attained by 43% of the patients, while 64% were discharged from the hospital with reduced psychiatric medication. Metabolic health showed improvement, and nearly all patients, with the exception of one, experienced significant weight loss [213]. In a 4-month single-arm pilot trial involving 23 patients with BD and SZ, KD therapy led to the reversal of metabolic syndrome. Participants with SZ demonstrated an average improvement of 32% according to the Brief Psychiatric Rating Scale. Additionally, 69% of the participants with BD exhibited more than a one-point improvement in the Clinical Global Impression (CGI) score. Overall, the pilot trial indicates dual metabolic–psychiatric benefits from the KD [10]. In another pilot trial lasting 6 to 8 weeks, applying the KD in BD, involving 27 recruited participants, demonstrated the feasibility and safety of this approach in a psychiatric population [214]. Preliminary results from the pilot trial showed increased ketone levels correlated with reduced impulsivity and anxiety [215]. Recently, several protocols have been published, introducing randomized controlled trials for the KD for refractory MDD and BD [208,209,210]. Before the contemporary era of new trials on the KD in mental illness, a series of studies in the pediatric domain demonstrated the potential to enhance core symptoms and improve the core features of autism spectrum disorder [216,217,218,219].

Despite evident limitations such as small sample sizes and the absence of a control group, the existing studies thus far have demonstrated the feasibility, tolerability, and significant improvements in psychiatric symptoms associated with the application of the KD in various serious, chronic, and refractory mental disorders. Moreover, it has been shown that the highly favorable safety profile known from a century of research in epilepsy and decades of research in obesity medicine is reproducible in these vulnerable cohorts. However, adequate real-world patient data are lacking. Further studies encompassing a broader range of disorders, particularly controlled trials, are urgently needed to establish conclusive evidence regarding the efficacy of the KD for various psychiatric conditions, including OCD.

## 10. Putative Mechanisms of the Ketogenic Diet in OCD

The KD primarily lowers blood glucose levels and promotes metabolic recovery but also exerts pleiotropic effects through fatty acids and ketone bodies, including anticonvulsant, anti-inflammatory, mitochondrial biogenesis modulation, and antioxidative properties. Additionally, by simulating the metabolic state of fasting, the KD influences hormone, neurotransmitter, and neuropeptide levels [220,221]. Evidence suggests that the KD’s multifaceted effects can be leveraged to address the complex pathophysiological dysfunctions underlying OCD, as described in detail above (Figure 1).

The KD compensates for deficiencies in cellular energy metabolism caused by reduced glucose uptake and inefficient glycolysis by significantly increasing the levels of circulating ketone bodies, thereby replacing glucose as the primary fuel source [222]. In the presence of oxygen, cellular energy is primarily derived from glucose metabolism, where glucose is converted to pyruvate through glycolysis, followed by oxidative phosphosphorylation in the mitochondria. In the absence of glucose, energy is produced through the breakdown of fatty acids and proteins. Fatty acid oxidation leads to the production of ketone bodies [223]. Switching to alternative primary energy sources, such as ketone bodies, could treat cerebral dysmetabolism and, thus, alleviate symptom burden [13]. Ketone bodies yield a greater amount of adenosine triphosphate (ATP) compared to glucose, earning them the nickname of a “super fuel”. For instance, 100 g of acetoacetate produces 9400 g of ATP, and 100 g of beta-hydroxybutyrate yields 10,500 g of ATP, while the same amount of glucose only generates 8700 g of ATP. This enhanced ATP production enables the body to maintain efficient fuel utilization even in the presence of a caloric deficit. The KD might thus ameliorate PFC dysfunction in OCD by optimizing energy metabolism.

Additionally, ketone bodies have been shown to reduce damage caused by free radicals and enhance the body’s capacity to counteract oxidative stress through increased antioxidant activity [224]. Ketone bodies also regulate mitochondrial functions and redox signaling through the induction of low redox signaling molecules. This process may ultimately increase the levels of antioxidants (e.g., glutathione (GSH)) and detoxification enzymes, thereby improving brain function and alleviating a wide range of neuropsychiatric symptoms [225,226]. Moreover, ketone bodies have been observed to promote the generation of new mitochondria (mitochondrial biogenesis) and decrease mitochondrial permeability [227]. Mitochondrial dysfunction resulting from neuronal injury leads to the production of reactive oxygen species (ROS), reactive electrophile species (RES), and reactive nitrogen species (RNS), which are linked to neuronal death and neurotoxicity in neurodegenerative diseases [228]. A study demonstrated that BHB plays a role in scavenging ROS and hydroxyl radicals, potentially mitigating oxidative damage [229]. Moreover, BHB preserved mitochondrial function and enhanced cell survival by directly reducing cellular ROS levels. Another study showed that the KD reduced the expression of the oxidative stress marker malondialdehyde in a murine model of multiple sclerosis (MS) [230]. The KD may thus effectively address mitochondrial dysfunction associated with OCD.

Accumulating evidence suggests that ketosis increases glutamate removal and reduces glutamate-induced excitability, which could lead to a further efficient glutamate exchange to GABA, increasing GABAergic activity and thus restoring the brain’s glutamate–GABA balance [231,232,233]. This neurochemical model may account for the effectiveness of ketosis in treating epilepsy and, given its shared pathophysiological characteristics with OCD [234], offers a compelling rationale for how ketosis could alleviate obsessive–compulsive symptoms.

AMP-activated protein kinase (AMPK) senses cellular energy levels, activating when energy is low to reduce ATP consumption. Additionally, AMPK regulates inflammation by activating Nuclear Factor (NF)-κB, which promotes the transcription of TNFα, IL-1β, and IL-6 [235]. The KD was found to reduce AMPK activation in a glaucoma mouse model and in mice subjected to kainic acid-induced hippocampal cell death, resulting in decreased expression of pro-inflammatory molecules [236,237]. In the CNS, the NLRP3 inflammasome is a key mediator of inflammatory signaling. BHB inhibits NLRP3 inflammasome activation by preventing ATP-induced ASC oligomerization and potassium efflux, which are essential for inflammasome assembly [237]. The KD’s potent anti-inflammatory properties may help combat dysregulated inflammation in OCD.

## 11. Conclusions and Suggestions for Future Research

Overall, the reviewed epidemiological, clinical, and animal studies suggest a converging link between altered insulin signaling, glucose dysmetabolism, dysregulated inflammatory responses, mitochondrial dysfunction, and OCD. This preliminary evidence highlights the often-overlooked importance of metabolic assessments in OCD patients. Clinicians should evaluate the familial risk of metabolic disorders and monitor metabolic parameters (e.g., body mass index, blood glucose and insulin levels, dietary habits, physical activity, and sleep patterns) throughout treatment, particularly when prescribing potentially obesogenic agents, to better understand and mitigate their potential impact on OCD severity and progression and general health. Controlled, well-designed clinical studies are required to evaluate the efficacy of the KD in managing OCD. A combined approach involving metabolic therapies such as nutrition, ketosis, insulin sensitizers, and potentially other agents to enhance mitochondrial and metabolic function could also offer a promising treatment strategy for OCD. Addressing metabolic dysfunction represents an unmet need, providing a roadmap for future research and advancing the field of metabolic psychiatry application in OCD. Future research should investigate controlled trials on the impact of the KD in treatment-resistant OCD, focusing on inflammation, mitochondrial function, and neurocircuitry abnormalities.

## Figures and Tables

**Figure 1 nutrients-17-00031-f001:**
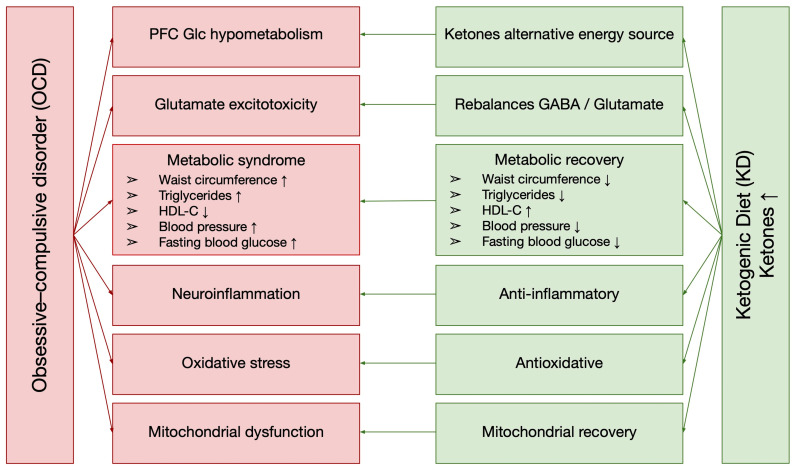
Metabolic and immunological alterations in OCD and the potential role of the KD in targeted treatment. Abbreviations: GABA, γ-aminobutyric acid; Glc, glucose; HDL-C, high-density lipoprotein cholesterol; ↑ indicates an increase, ↓ indicates a decrease.

**Table 1 nutrients-17-00031-t001:** OCD symptom dimensions.

OCD Dimension	Obsessions	Compulsions
Contamination/Cleaning	Fear of contamination, disgust-related thoughts	Excessive washing, cleaning
Symmetry/Ordering/Arranging/Counting	Need for symmetry, exactness, or “just right” feelings	Arranging, ordering, counting
Sexual/Religious	Intrusive sexual or blasphemous thoughts	Mental rituals, reassurance seeking
Aggression	Fear of harming others, violent images	Checking, avoidance behaviors
Somatic	Preoccupation with illness or bodily functions	Checking health-related cues, body-focused rituals
Hoarding/Collecting	Fear of discarding items, attachments to objects	Accumulating, refusing to discard possessions
Miscellaneous	Varied intrusive thoughts or images	Diverse rituals, non-categorized behaviors

**Table 2 nutrients-17-00031-t002:** Criteria for the diagnosis of metabolic syndrome according to the American Heart Association and the National Heart, Lung, and Blood Institute (AHA/NHLBI) and the International Diabetes Federation (IDF).

Risk Factors	AHA/NHLBI	IDF
Assessment of Components	≥3 of the following risk factors	≥3 of the following risk factors
Waist circumference	Men > 102 cm, women > 88 cm	Men > 94 cm, women > 80 cm
Triglycerides	≥150 mg/dL or lipid-lowering drugs	>150 mg/dL (1.7 mmol/L)
HDL cholesterol	Men < 40 mg/dL, women < 50 mg/dL or medication	Men < 40 mg/dL (1.03 mmol/L), women < 50 mg/dL (1.29 mmol/L)
Blood pressure	≥130 mmHg SBP, ≥85 mmHg DBP or antihypertensives	>130 mmHg, >85 mmHg
Fasting blood glucose	≥100 mg/dL or anti-diabetics	>100 mg/dL (5.6 mmol/L)

## Data Availability

No new data were created or analyzed in this study. Data sharing is not applicable.

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
