# Peer review of "Ketogenic Diet as a Nutritional Metabolic Intervention for Obsessive–Compulsive Disorder: A Narrative Review"

_nutrients, 2024, doi:10.3390/nu17010031_

Round 1

Reviewer 1 Report

Comments and Suggestions for Authors

Lounici et al review the research field of the OCD in the context of putative therapeutic utility of ketogenic diet. The paper was well written and easy to read. The authors provided good overview of OCD symptoms, brain activation and molecular abnormalities. The rationale for KD nutritional intervention in OCD was presented. The paper should significantly contribute to the field, below are some minor comments.

1.       P. 6. Authors write: “Additionally, hypofunction in the OFC and ACC may hinder the learning of new associations between cues, actions, and outcomes, undermining the effectiveness of exposure therapy. These findings suggest that shifting to alternative primary energy sources, such as ketone bodies, may help address cerebral dysmetabolism and, as a result, reduce symptom burden [13].”

What is the rationale for this suggestion, why authors think that modification of the nutritional energy substrate will normalize hypofunction in one brain area but hyperfunction in another, such as an increased lateralization in OCD.

2.       P. 8. Authors write: “To exert these functions, insulin crosses the blood-brain barrier and binds to specific insulin receptors on neurons and glial cells [129]”.

The role of brain-derived insulin in the brain function cannot be excluded, for review see Dakic T, et al The Expression of Insulin in the Central Nervous System: What Have We Learned So Far? Int J Mol Sci 24, 2023.

3.       P.10. Authors write: “Ketosis is a phylogenetically old physiological condition characterized by increased levels of circulating ketone bodies. Ketone bodies are produced in a process termed keto-genesis, which occurs in the mitochondrial matrix of hepatocytes [179].

I would rather say that ketogenesis and not ketosis is a phylogenetically old process.”

4.       P.10. Authors say: “Ketosis has a long track record in human history.” How was that documented? Data from 1920 is rather modern.

5.       After reading the paper, I understood the KD should be good for all psychiatric disorders, OCD is not an exception by boosting neuronal energy via ketonic bodies. Do authors think that certain psychiatric diseases should be more sensible that others for such nutritional intervention.

Author Response

Dr. Timur Liwinski
University Psychiatric Clinics Basel
Wilhelm Klein-Strasse 27, 4002 Basel, Switzerland

Subject: Response to Reviewer Comments and Revised Manuscript Submission – “Ketogenic Diet as a Nutritional Metabolic Intervention for Obsessive-Compulsive Disorder: A Narrative Review”

Dear Editor, 

we are grateful for the opportunity to address the reviewers' insightful critiques and to submit our revised manuscript. We have thoroughly reviewed and integrated all suggested changes, recognizing their importance in strengthening the scientific and clinical value of our manuscript. Requested revisions have been made, and we have provided a detailed response to each reviewer comment, with the corresponding changes highlighted. 

Kind regards,

Timur Liwinski
 1.    P. 6. Authors write: “Additionally, hypofunction in the OFC and ACC may hinder the learning of new associations between cues, actions, and outcomes, undermining the effectiveness of exposure therapy. These findings suggest that shifting to alternative primary energy sources, such as ketone bodies, may help address cerebral dysmetabolism, improve learning, and as a result, reduce symptom burden [13].”

What is the rationale for this suggestion, why authors think that modification of the nutritional energy substrate will normalize hypofunction in one brain area but hyperfunction in another, such as an increased lateralization in OCD.

Answer: Thank you for the important comment. Several studies showed OCD patients have cerebral glucose metabolism dysregulation (Cerebral Glucose Metabolism in Obsessive-Compulsive Hoarding | American Journal of Psychiatry, A study of brain glucose metabolism in obsessive-compulsive disorder patients - PubMed). The authors suggest that modification of the nutritional energy substrate may address cerebral glucose dysmetabolism and potentially correct the hypo/hyperfunction observed in OCD. In attempt to make this clearer, we have adjusted the language to the following: These findings suggest that shifting to alternative primary energy sources, such as ketone bodies, may help address cerebral dysmetabolism, improve learning, and as a result, reduce symptom burden [13].” (page 5, line 266 ff.)

2.     P. 8. Authors write: “To exert these functions, insulin crosses the blood-brain barrier and binds to specific insulin receptors on neurons and glial cells [129]”.
The role of brain-derived insulin in the brain function cannot be excluded, for review see Dakic T, et al The Expression of Insulin in the Central Nervous System: What Have We Learned So Far? Int J Mol Sci 24, 2023.

Answer: Thank you for pointing this out. Although it is still unknown which CNS insulin roles can be attributed to peripheral pancreas-secreted insulin and which ones to the brain-derived insulin (The Expression of Insulin in the Central Nervous System: What Have We Learned So Far? - PMC), studies on neurodegenerative diseases often associate Alzheimer’s disease with insulin-deficient and/or insulin-resistant brain states. Impaired transport of pancreatic insulin across the BBB is associated with impaired proper functioning of brain-derived insulin (Cerebrovascular insulin receptors are defective in Alzheimer’s disease - PMC).

3.    P.10. Authors write: “Ketogenesis is a phylogenetically old physiological condition characterized by increased levels of circulating ketone bodies. Ketone bodies are produced in a process termed keto-genesis, which occurs in the mitochondrial matrix of hepatocytes [179].
I would rather say that ketogenesis and not ketosis is a phylogenetically old process.”
Answer: We agree, ketogenesis is a better term to use. Thank you for the suggestion and we have corrected this.

4.    P.10. Authors say: “Ketosis has a long track record in human history.” How was that documented? Data from 1920 is rather modern. 

Answer: Thank you for your comment. We acknowledge that our initial wording was unclear. We have rephrased the sentence to clarify that we are referring to the evolutionary natural history rather than the written historical record:  

“Ketosis represents a phylogenetically conserved metabolic state with a long-standing role in human evolutionary history, underscoring its significance as an adaptive mechanism.” (page 10, line 617)

5.    After reading the paper, I understood the KD should be good for all psychiatric disorders, OCD is not an exception by boosting neuronal energy via ketonic bodies. Do authors think that certain psychiatric diseases should be more sensible that others for such nutritional intervention.

Answer: Thank you for your thoughtful question. We have added the following passage to address it:
“While the validity of discrete psychiatric diagnoses has been widely debated, the framework remains intact in both the DSM-5 and the upcoming ICD-11 classification systems. Diagnosis-specific research continues to be prioritized within psychiatry. Although overlapping pathophysiological features, such as those described in the review, have been observed across various psychiatric disorders, it remains valuable to provide a comprehensive biological treatment rationale for specific diagnoses. This approach not only facilitates targeted research efforts but also supports the development of both diagnosis-specific and transdiagnostic clinical studies.” (page 14, line 823 ff.)

Reviewer 2 Report

Comments and Suggestions for Authors

Thank you for giving me the opportunity to review this manuscript.

This narrative review described what kind of biological factors may affect OCD.

1) Please describe that this study was a narrative review in the title.

2) This manuscript described what kind of biological factors may affect OCD compared with healthy population. However, the diagnostic challenges in psychiatry are that the validity of diagnosis is very low although the reliability is high. This means that those biological factors the authors mentioned do not distinguish OCD from other mental disorders. In other words, those factors are not always OCD specific. I think the biological factors will not improve OCD because they are not OCD specific. Please describe how these problems affect those results and what kind of studies are warranted.

I think it is better to review this manuscript.

Author Response

Dr. Timur Liwinski
University Psychiatric Clinics Basel
Wilhelm Klein-Strasse 27, 4002 Basel, Switzerland

Subject: Response to Reviewer Comments and Revised Manuscript Submission – “Ketogenic Diet as a Nutritional Metabolic Intervention for Obsessive-Compulsive Disorder: A Narrative Review”

Dear Editor, 

we are grateful for the opportunity to address the reviewers' insightful critiques and to submit our revised manuscript. We have thoroughly reviewed and integrated all suggested changes, recognizing their importance in strengthening the scientific and clinical value of our manuscript. Requested revisions have been made, and we have provided a detailed response to each reviewer comment, with the corresponding changes highlighted. 

Kind regards,

Timur Liwinski
 1) Please describe that this study was a narrative review in the title.

Thank you for the suggestion, the proposed new title is: Ketogenic Diet as a Nutritional Metabolic Intervention for Obsessive-Compulsive Disorder: A Narrative Review.

2) This manuscript described what kind of biological factors may affect OCD compared with healthy population. However, the diagnostic challenges in psychiatry are that the validity of diagnosis is very low although the reliability is high. This means that those biological factors the authors mentioned do not distinguish OCD from other mental disorders. In other words, those factors are not always OCD specific. I think the biological factors will not improve OCD because they are not OCD specific. Please describe how these problems affect those results and what kind of studies are warranted.
Answer: Thank you for your thoughtful suggestion. We have added the following passage to address it:
“While the validity of discrete psychiatric diagnoses has been widely debated, the framework remains intact in both the DSM-5 and the upcoming ICD-11 classification systems. Diagnosis-specific research continues to be prioritized within psychiatry. Although overlapping pathophysiological features, such as those described in the review, have been observed across various psychiatric disorders, it remains valuable to provide a comprehensive biological treatment rationale for specific diagnoses. This approach not only facilitates targeted research efforts but also supports the development of both diagnosis-specific and transdiagnostic clinical studies.” (page 14, line 823 ff.)